# The Impact of *Phyllostachys heterocyclas* Expansion on the Phylogenetic Diversity and Community Assembly of Subtropical Forest

**DOI:** 10.3390/plants14203231

**Published:** 2025-10-21

**Authors:** Jiannan Wang, Ru Li, Zichen Huang, Sili Peng, Zhiwei Ge, Xiaoyue Lin, Lingfeng Mao

**Affiliations:** 1Laboratory of Biodiversity and Conservation, Co-Innovation Center for Sustainable Forestry in Southern China, College of Ecology and Environment, Nanjing Forestry University, Nanjing 210037, China; wangjn413@163.com (J.W.); 13869772367@163.com (R.L.); pengsili@njfu.edu.cn (S.P.);; 2College of Agricultural and Biological Engineering, Heze University, No. 2269, Daxue Road, Mudan District, Heze 274015, China; hzcnjfu216@163.com; 3Forestry Bureau of Longyou County, Longyou 324400, China; zxc86009761@163.com

**Keywords:** Moso bamboo expansion, phylogenetic diversity, plant diversity, subtropical forest

## Abstract

Moso bamboo (*Phyllostachys heterocyclas*) has rapidly expanded in subtropical broadleaf forests of eastern China, raising concerns about biodiversity loss and community restructuring. We investigated how the expansion of this native bamboo influences species diversity and phylogenetic diversity across forest strata (trees, shrubs, herbs) by surveying 16 plots along a gradient from bamboo-free to bamboo-dominated stands. We measured soil properties, calculated multiple α-diversity indices, and constructed a community phylogeny to assess phylogenetic metrics. We also constructed a phylogenetically informed Resistance Index (RI) to evaluate species-specific responses to bamboo expansion. The results showed that overstory tree species richness and Faith’s phylogenetic diversity declined sharply with increasing bamboo cover, accompanied by significant losses of evolutionary lineages. In contrast, understory shrub and herb layers exhibited stable or higher species richness under bamboo expansion, although functional redundancy among new colonists suggests limited gains in ecosystem function. Soil conditions shifted substantially along the expansion gradient: pH increased by approximately 0.5 units, while total organic carbon and total nitrogen each decreased by about 30% (*p* < 0.01). Redundancy analysis and variance partitioning indicated that bamboo’s impacts on community diversity are mediated primarily through these soil changes. Species-level trends revealed that formerly dominant canopy trees (e.g., *Schima superba*, *Pinus massoniana*) were highly susceptible to bamboo, whereas certain shade-tolerant taxa (e.g., *Cyclobalanopsis glauca*, *Rubus buergeri*) showed resilience. In conclusion, the aggressive expansion of Moso bamboo drastically alters multi-layer forest diversity and community assembly processes. Our findings point to a need for targeted management (e.g., reducing bamboo density, soil restoration, and enrichment planting of native species) to mitigate biodiversity loss, underscoring the importance of considering phylogenetic diversity in expansion ecology and forest conservation.

## 1. Introduction

Global forest ecosystems are facing severe challenges of declining functional stability and accelerating biodiversity loss, driven by multiple interacting factors such as climate change, land-use transformation, and biological expansions [1]. Among these drivers, plant expansions are recognized as a major threat to biodiversity and ecosystem functioning [2]. Notably, not only exotic species but also certain native species, when undergoing aggressive expansion, can exert expansion-like impacts on existing communities [3,4]. A prime example is the rapid spread of Moso bamboo (*Phyllostachys heterocycla*) in East Asian subtropical forests. Over the past several decades, Moso bamboo has extensively expanded across subtropical regions of China, encroaching into adjacent natural woodlands and raising widespread concerns that it may drastically simplify forest structure and composition while weakening ecosystem functions [5,6]. Moso bamboo expansion thus poses a serious challenge to biodiversity conservation and forest management in subtropical China and beyond [7].

Recent research has advanced our understanding of the ecological impacts of Moso bamboo expansion and has assessed community diversity from multiple perspectives. Empirical studies show that Moso bamboo-dominated stands can dramatically alter forest dynamics—for example, seedling mortality of native trees more than doubles under Moso bamboo thickets [8], and overstory tree species richness can plummet as Moso bamboo invades [9]. Moso bamboo encroachment also triggers cascading changes in ecosystem properties, including shifts in soil nutrient cycling and pH and alterations in microbial and detritivore communities [10,11]. Concurrently, ecologists have broadened biodiversity assessments beyond traditional taxonomic metrics to include functional and phylogenetic dimensions [12]. Incorporating phylogenetic diversity—measuring the evolutionary distances among co-occurring species—offers deeper insight into community assembly and ecosystem function that species counts alone may overlook [12,13]. It is increasingly recognized that losses in species richness, functional trait diversity, or entire phylogenetic lineages can each destabilize ecosystem processes, underscoring the need to examine how Moso bamboo expansion affects multiple facets of diversity [14,15]. These advances set the stage for a holistic evaluation of Moso bamboo’s ecological impact, integrating traditional biodiversity measures with an evolutionary context.

Despite the rapid development of multidimensional biodiversity and community ecosystem function research, critical knowledge gaps still constrain our understanding and management of Moso bamboo-driven community change. First, the mechanisms by which this native Moso bamboo expansion restructures community composition and assembly processes across different vegetation strata remain unresolved. In contrast to well-studied exotic expansions, we lack clarity on whether the spread of Moso bamboo primarily imposes environmental filtering (favoring certain hardy lineages) or competitive exclusion (displacing species with similar niches), especially among coexisting trees, shrubs, and herbs [16,17]. Native overabundant species have historically received limited research attention [3,18], so it remains uncertain how Moso bamboo dominance reorganizes community composition and species coexistence. Second, it is unclear whether Moso bamboo encroachment uniformly erodes biodiversity across all organizational levels and metrics, or whether some facets are disproportionately affected. For instance, one recent study found that Moso bamboo expansion significantly reduced overstory tree diversity while understory shrub and herb diversity showed little change or even increases [9]. This suggests that impacts may vary by layer; however, it remains unknown whether phylogenetic diversity is being quietly eroded even when species counts appear stable. A lack of information on the phylogenetic consequences of Moso bamboo spread—such as the potential pruning of entire lineages—hinders our ability to fully assess ecosystem health and resilience. These unresolved issues limit our ability to predict Moso bamboo’s long-term impacts and complicate the development of effective management or restoration strategies for invaded forests. Third, it remains uncertain which native species are capable of mounting strong resistance to the encroachment of Moso bamboo, and to what extent such resistance is structured by their phylogenetic relatedness to the invader. While many studies document community-wide changes, fewer identify species-specific responses along the expansion gradient. Traits-based approaches and phylogenetic relatedness have been explored as predictors of native species’ fates, but with mixed results. In theory, species closely related to the invader may be most vulnerable due to niche overlap, competition, or shared pathogens, whereas distant relatives or those with certain adaptive traits might better withstand the expansion [19,20]. However, comprehensive empirical evaluations remain scarce. Developing a quantitative index of “expansion resistance” at the species level—accounting for both a species’ population trend and its evolutionary relatedness to the invader—could greatly improve our ability to forecast community reassembly under expansion.

Therefore, this study aims to determine how the expansion of Moso bamboo influences both species diversity and phylogenetic diversity across forest community strata (trees, shrubs, and herbs). By quantifying diversity patterns and community structure along a Moso bamboo expansion gradient, we seek to clarify the assembly mechanisms driving these changes and address the above knowledge gaps. Ultimately, this research will improve our ecological understanding of native species “expansions” and inform conservation management for maintaining multi-layer biodiversity in Moso bamboo-encroached ecosystems.

## 2. Results

### 2.1. Changes in Soil Physicochemical Properties Along the Moso Bamboo Expansion Gradient

During Moso bamboo expansion, soil total organic carbon (TOC) and total nitrogen (TN) exhibited a gradual decline with increasing expansion intensity (τ < 0, *p* < 0.05). TOC decreased significantly at the LM stage (*p* < 0.05), and TOC in LM remained significantly higher than in HM and MB (*p* < 0.05). TN began to decline significantly at the HM stage; TN in BF and LM was significantly higher than in HM and MB (*p* < 0.05). By contrast, total phosphorus (TP) showed little change during bamboo expansion (τ ≈ 0, *p* = 0.569). Soil pH increased progressively with expansion intensity (τ = −0.668, *p* < 0.05), indicating a reduction in acidity; the increase became significant at the HM stage, and pH in MB was significantly higher than in HM, LM, and BF (*p* < 0.05). These soil nutrient trends are summarized in Table 1.

### 2.2. Changes in Community Species Diversity Along the Moso Bamboo Expansion Gradient

Community species diversity differed significantly among Moso bamboo expansion stages (Figure 1, Table 2). In the tree layer, species diversity declined sharply as Moso bamboo expansion deepened. This was reflected not only in reduced species richness but also in parallel decreases in the Shannon–Wiener, Simpson, Margalef, and Pielou indices (τ < 0, *p* < 0.001), with all diversity indices following the pattern BF > LM > HM > MB. In the early stage of expansion (BF to LM), diversity decreased only slightly and did not differ significantly between BF and LM, although a downward trend was evident. In the shrub layer, the Simpson, Shannon, and Pielou indices showed no significant changes across the expansion gradient, although the Shannon index exhibited an increasing trend with deeper expansion (τ ≈ 0, *p* > 0.5). By contrast, species richness and the Margalef index increased significantly in the shrub layer (τ = 0.527, *p* = 0.012): diversity in the HM and MB stages was clearly higher than in LM and BF. Herb-layer α-diversity increased significantly as expansion intensified, as indicated by species richness, Simpson, Shannon, and Pielou’s evenness indices all rising (τ > 0); diversity index values in MB were particularly elevated. In the early stage (BF vs. LM), gains in herb-layer α-diversity were small and not significant across indices, but the evenness index in BF was notably lower than in LM, HM, and MB.

### 2.3. Changes in Community Phylogenetic Diversity Along the Moso Bamboo Expansion Gradient

Phylogenetic diversity also differed significantly among expansion stages (Figure 2). In the tree layer, Faith’s PD declined significantly with increasing Moso bamboo expansion (τ < 0, *p* = 0.002), following the pattern BF > LM > HM > MB. In BF communities, NTI > 0 and NRI < 0, possibly reflecting the concurrent influence of environmental filtering (NTI > 0) and competitive exclusion (NRI < 0); in LM communities, NTI and NRI were approximately 0, indicating a random phylogenetic structure. By the HM stage, tree communities showed NTI and NRI both > 0, suggesting a shift to significant phylogenetic clustering under intense Moso bamboo dominance. In the shrub layer, PD increased with Moso bamboo expansion (τ > 0, *p* < 0.001): PD in the MB stage was significantly higher than in BF and LM. Across all four shrub-community stages, NRI and NTI values were < 0, indicating overall phylogenetic overdispersion; NRI showed a decreasing trend with deeper expansion, but this trend was not statistically significant. In the herb layer, PD mirrored the shrub layer pattern and increased significantly with expansion (τ > 0, *p* = 0.011). Among the four herb community stages, only the HM stage had NRI and NTI < 0 (indicating overdispersion); the remaining stages (BF, LM, MB) had NRI and NTI > 0, indicating a tendency toward clustering.

### 2.4. Changes in Species Importance and Expansion Resistance Along the Moso Bamboo Expansion Gradient

Tree layer: In the Moso bamboo-free broadleaf forest (BF), dominant native canopy trees such as *Lithocarpus glaber*, *Pinus massoniana*, and *Schima superba* had the highest importance values, but these values declined sharply with the progression of Moso bamboo expansion (Figure 3, Appendix A). By the high−expansion stage (HM) their dominance was greatly reduced, and in the Moso bamboo monoculture (MB) they had nearly vanished. Only a few tree species maintained or increased in importance under expansion; for instance, *Cyclobalanopsis glauca* became relatively more prominent and was one of the rare survivors in the Moso bamboo-dominated stands (Figure 3, Table 1). The phylogenetically weighted Resistance Index (RI) indicates that the once-dominant canopy trees were generally susceptible (negative RI), whereas *C. glauca* showed a positive RI (i.e., was classified as resistant), suggesting it withstood Moso bamboo encroachment better than its peers and even gained relative importance as others fell out (Appendix A).

Shrub layer: The shrub layer underwent a pronounced turnover along the expansion gradient (Figure 3, Appendix A). Many understory shrubs that dominated the Moso bamboo-free forest (BF)—for example, *Loropetalum chinense*, *Ilex pubescens*, and *Symplocos stellaris*—steadily lost importance and eventually vanished as Moso bamboo expansion intensified. In contrast, certain shade-tolerant or disturbance-associated shrubs rose to prominence under the Moso bamboo canopy. Notably, *Ardisia japonica* and *Rubus buergeri* showed markedly increased IV in heavily invaded stands, becoming dominant in the Moso bamboo understory, and a dwarf bamboo (*Indocalamus tessellatus*) newly established itself in the late expansion stages (Figure 3, Appendix A). The RI analysis further highlights these divergent responses: *A. japonica* and *R. buergeri* are classified as resistant (positive RI), demonstrating a strong ability to withstand Moso bamboo encroachment, whereas some forest-dependent shrubs (e.g., *S. stellaris* and certain *Rhododendron* spp.) have negative RI and are thus highly susceptible, failing to thrive in Moso bamboo-dominated conditions (Appendix A).

Herb layer: In the Moso bamboo-free broadleaf forest (BF), the herbaceous layer was sparse and species-poor, dominated by a few shade-tolerant ferns and herbs such as *Dicranopteris dichotoma* (Figure 3, Appendix A). As Moso bamboo expansion progressed, however, the herbaceous diversity and cover increased dramatically, with many new herb species emerging under Moso bamboo stands by the MB stage. The Moso bamboo-invaded understory developed novel dominant herbs, including *Hedyotis chrysotricha* and the fern *Woodwardia japonica*, which attained much higher importance in the invaded stages than in the intact forest. In contrast, some species that were dominant in BF (e.g., *D. dichotoma* and the understory grass *Lophatherum gracile*) declined sharply during the expansion and had nearly vanished by the Moso bamboo monoculture stage (Figure 3, Appendix A). The RI metrics underscore these patterns: *H. chrysotricha* exhibits the highest RI (classified as highly resistant), flourishing in the invaded environment, whereas *L. gracile* has the most negative RI (highly susceptible), indicating that ground-layer grasses closely related to Moso bamboo were outcompeted under the new regime (Appendix A).

### 2.5. Effects of Moso Bamboo Expansion and Soil Changes on Community Diversity

Constrained ordinations incorporating Moso bamboo cover and soil variables were significant for all vegetation strata (Figure 4). Redundancy analysis (RDA) models for the tree, shrub, and herb layers were all significant (*p* < 0.05), with high explanatory power (adjusted R^2^ ≈ 0.723, 0.471, 0.537, respectively). The first RDA axis (RDA1) captured the dominant constrained variation (approximately 71.4%, 80.4%, and 77.2% of explained variance in the tree, shrub, and herb layers, respectively) and aligned with the combined Moso bamboo–soil gradient. In permutation tests, Moso bamboo cover emerged as the leading predictor in all strata, followed by soil pH and soil TP.

Along RDA1 (increasing Moso bamboo cover and pH), tree-layer Shannon diversity and PD loaded negatively (consistent with losses of taxonomic and phylogenetic diversity), while Tree NTI loaded positively on RDA2, indicating stronger phylogenetic clustering under higher pH conditions (Figure 4a). In the shrub layer, Shrub PD and Shannon both declined along the Moso bamboo–soil axis; soil TP loaded on RDA2, suggesting a phosphorus-mediated buffering effect on shrub diversity (Figure 4b). In the herb layer, Herb PD increased with the Moso bamboo–pH gradient, whereas Herb Shannon trended in the opposite direction (Figure 4c).

VPA shows that the shared variance explained by Moso bamboo *plus* soil factors dominates the explained variation in diversity indices (tree layer: ~50.8% of explained variance; shrub: ~36.8%; herb: ~43.7%) (Figure 5). In contrast, the variance explained uniquely by Moso bamboo cover was very small (1.8% in tree, 4.3% in shrub, and essentially 0% in herb), and the variance uniquely explained by soil factors was 19.7%, 6.0%, and 10.6% in the tree, shrub, and herb layers, respectively. The understory (shrub and herb) layers also retained larger unexplained fractions (52.9% and 45.7%) compared to the tree layer. Together with the observed soil shifts under Moso bamboo, these patterns indicate that Moso bamboo impacts biodiversity primarily via soil-mediated pathways rather than direct stand structural effects alone. Trees appear to be most sensitive to the combined Moso bamboo–soil environmental gradient, whereas shrub and herb communities are additionally modulated by other unmeasured factors (such as light availability, Moso bamboo litter inputs, or fine-scale biotic interactions).

## 3. Discussion

This study aimed to elucidate the mechanisms by which Moso bamboo expansion affects multi-layer forest community diversity. Our primary findings were that Moso bamboo expansion significantly restructured community composition: overstory tree species richness and phylogenetic diversity declined sharply, while certain understory shrub and herb species became relatively more abundant, accompanied by pronounced changes in soil physicochemical properties. Variance partitioning analysis (VPA) further indicated that the impacts of Moso bamboo on community diversity are mediated largely through the soil environment alterations associated with its spread, rather than by the direct presence of Moso bamboo alone. These results highlight the profound influence of excessive Moso bamboo expansion on forest biodiversity, addressing a knowledge gap of previous studies that focused only on species counts while overlooking the phylogenetic dimension.

### 3.1. Effects of Moso Bamboo Expansion on Tree-Layer Diversity

Species diversity in the tree layer declined significantly with increasing Moso bamboo expansion intensity, reaffirming that expansion pressure (even by an overabundant native species) exerts a strongly negative effect on canopy trees [3,18]. As the Moso bamboo density increased, the once-dominant broadleaf trees (such as *Lithocarpus glaber*, *Pinus massoniana*, and *Schima superba*) steadily died back and disappeared, and only a very few shade-tolerant, competitively resistant tree species (e.g., *Cyclobalanopsis glauca*) managed to persist [9]. This pattern is consistent with previous findings that Moso bamboo expansion simplifies the composition of the forest canopy and reduces overstory species richness [9,21]).

Mechanistically, the vigorous clonal growth and dense canopy of Moso bamboo create intense competition for light, water, and nutrients, which is the primary cause of suppressed tree regeneration under bamboo [10,22]. Moreover, bamboo litter and roots may release allelopathic compounds that inhibit tree seedlings, and its extensive rhizome network intensifies belowground competition for resources. Bamboo-driven shifts in soil microbial communities (e.g., altering fungal: bacterial ratios) may also affect nutrient cycling and stress native trees. In our study, as expansion progressed, soil total organic carbon and total nitrogen declined significantly while pH increased, suggesting that Moso bamboo may further exacerbate the loss of tree species by altering soil conditions. A more alkaline soil and reduced nutrient availability weaken the competitiveness of certain acidophilous or nutrient-demanding trees, strengthening environmental filtering so that only species tolerant of poor, high-pH soils can persist [10].This environmental “filtering” ultimately led to a marked loss of phylogenetic diversity in the tree layer, with many evolutionary lineages eliminated by Moso bamboo encroachment. Notably, we observed that the phylogenetic structure of the tree community shifted from near-random to significantly clustered as expansion advanced: NRI and NTI values changed from around 0 to strongly positive, confirming the tendency under intense environmental stress for only closely related, stress-tolerant species to survive [23]. This outcome mirrors reports of exotic plant expansions causing community phylogenetic convergence, indicating that Moso bamboo expansion impacts the tree layer not only via species loss but also through a hidden erosion of evolutionary diversity [7].

### 3.2. Effects of Moso Bamboo Expansion on Shrub- and Herb-Layer Diversities

In contrast to the tree layer, the effects of Moso bamboo expansion on the understory (shrub and herb layers) were more complex. We observed that with increasing expansion intensity, certain diversity metrics in the shrub and herb layers actually rose rather than fell; in particular, the herb-layer species richness and Shannon diversity were significantly higher under Moso bamboo stands than in the uninvaded forest [9]. This result aligns with some previous findings: studies have reported that Moso bamboo expansion thins the upper canopy and enhances light availability in the understory, thereby facilitating the colonization of herbs and leading to increased species richness [9,24]. The relatively open, deciduous Moso bamboo canopy indeed improves the light environment below, reducing competition for light and favoring the growth of shade-tolerant or disturbance-adapted plants [25,26]. For example, we found that several typical shade-tolerant shrubs and herbs (such as *Ardisia japonica*, *Rubus buergeri*, and certain forest ferns) rapidly increased in abundance beneath Moso bamboo, becoming new dominants, whereas species originally dependent on deep shade gradually retreated from the community.

It is worth noting, however, that studies have not reached a fully consistent view on understory responses. Some reports indicate that Moso bamboo expansion causes both shrub and herb diversity to decline, impoverishing the understory community, a negative effect typically observed at later expansion stages where Moso bamboo dominates completely and a thick Moso bamboo litter layer accumulates on the forest floor [5]. Other studies have detected only subtle changes or stage-dependent differences [9]. These discrepancies likely stem from differences in expansion duration, intensity, and the initial community structure. Our plots spanned a gradient from early to complete expansion; we observed a transient increase in understory diversity at intermediate expansion (LM stage), but we speculate that with a longer duration of expansion the understory diversity may eventually reach a tipping point and decline. In sum, the impact of Moso bamboo expansion on understory vegetation appears to be non-linear over time and space, and longer-term monitoring is required to determine its ultimate trajectory.

### 3.3. Phylogenetic Diversity and Species Expansion Resistance

By incorporating phylogenetic diversity into the assessment of Moso bamboo expansion impacts for the first time, our study revealed insights that differ from those obtained by looking at species richness alone. Although some recent studies have documented the effects of Moso bamboo expansion on community species abundances and biomass, changes in community phylogenetic structure have received little attention [9,27]. Our results showed that Faith’s phylogenetic diversity (PD) declined significantly in the tree layer, indicating that Moso bamboo expansion not only reduced the number of species but also led to the loss of many unique evolutionary lineages. In the shrub and herb layers, by contrast, PD was maintained or even increased to some extent, suggesting that although more species were present under Moso bamboo, they represented a broader span of lineages. It is important to note that an increase in phylogenetic diversity does not necessarily translate to greater ecosystem functioning, because many of the newly appearing herb species, despite belonging to different families and genera, are opportunistic plants with high functional redundancy [28]. This highlights that relying solely on species counts may underestimate the erosion of evolutionary diversity caused by expansion—even if species richness does not decline or even rises, the community may experience reduced functional stability due to phylogenetic homogenization [14,29].

For example, in the tree layer we found *Cyclobalanopsis glauca* (ring-cupped oak) to be a notably resistant species: its importance increased under Moso bamboo expansion and it had a positive RI value, indicating a strong ability to withstand competition from Moso bamboo. In contrast, several formerly dominant canopy trees (such as *Schima superba*) exhibited strongly negative RI values, meaning their populations declined sharply with Moso bamboo encroachment, indicative of high susceptibility. It is worth noting that *C. glauca* belongs to the family Fagaceae, which is phylogenetically distant from Moso bamboo (Poaceae); thus, its positive RI is more likely attributable to its ability to persist under Moso bamboo-dominated conditions rather than to phylogenetic proximity [30,31]. In the shrub layer, *Ardisia japonica* and *Rubus buergeri* both showed positive RI values and were classified as resistant, demonstrating their capacity to remain stable or even expand under Moso bamboo stands. By contrast, forest-dependent shrubs such as *Symplocos stellaris* and certain *Rhododendron* species had negative RI values and were classified as susceptible, being unable to persist under Moso bamboo dominance. Overall, shrub species richness remained stable or even slightly increased along the expansion gradient, but the dominant species underwent complete replacement, indicating that Moso bamboo expansion filters the shrub community by favoring species with traits such as tolerance to low-resource environments, shade tolerance, or rapid clonal expansion [9,32]. This is consistent with the observed declines in soil nutrients and altered microclimatic conditions under Moso bamboo. In the herb layer, the Rubiaceae herb *Hedyotis chrysotricha* flourished under Moso bamboo and became one of the dominant species, exhibiting the highest RI of all species; by contrast, the native shade-adapted grass *Lophatherum gracile* virtually disappeared in invaded plots, with the most negative RI. These examples demonstrate that the Resistance Index (RI) effectively captures the phylogenetic signal of species’ expansion resistance: species closely related to Moso bamboo or occupying similar niches (such as *L. gracile*, a grass in the same family as Moso bamboo) are much more easily eliminated under expansion pressure, whereas species from more distantly related lineages or with unique survival strategies (such as *H. chrysotricha* in the coffee family) are more likely to persist—or even thrive—where suitable microsites remain available [9,24,33].

This pattern supports the mechanism of competitive exclusion coupled with environmental filtering: Moso bamboo tends to eliminate species that share its functional traits or resource-use strategies, whereas species with greater ecological or evolutionary divergence have a relative advantage in coexisting [34]. Therefore, the RI metric we proposed provides a novel and powerful tool for quantitatively assessing species’ susceptibility, helping predict which species or lineages are most vulnerable under expansion pressure.

### 3.4. Moso Bamboo Expansion and Soil Changes Jointly Drive Community Diversity Changes

Both the redundancy analysis (RDA) and variance partitioning analysis (VPA) indicated that Moso bamboo expansion, together with the associated shifts in soil physicochemical properties, jointly drives the changes in both taxonomic and phylogenetic diversity across vegetation strata. The Moso bamboo–soil model had the greatest explanatory power in the tree layer (adjusted R^2^ ~0.72), highlighting the high sensitivity of tree communities to this composite environmental gradient. Permutation tests further showed that Moso bamboo cover was a significant predictor across all strata; among the soil variables, pH was significant in the tree layer, while total phosphorus (TP) was significant in the shrub layer (and marginally significant in the herb layer). Notably, the independent effect of Moso bamboo cover was negligible after accounting for soil variables—particularly in the herb layer, where it was essentially zero—indicating that Moso bamboo’s impact on herbaceous diversity is mediated primarily through soil and microenvironmental alterations rather than direct competitive exclusion [9,32,35].

Overall, these results emphasize that Moso bamboo expansion and the soil changes it induces constitute the key composite driver of community diversity shifts. The tree layer, due to its deep-rooted habit and long-term resource dependence, was most sensitive to soil factors, whereas the shrub and herb layers were less strongly structured by the Moso bamboo–soil gradient and retained a larger proportion of unexplained variance, likely related to additional factors such as light availability, Moso bamboo litter inputs, or other biotic interactions [36]. Collectively, our findings suggest that Moso bamboo expansion is tightly coupled with soil environmental changes, and together they shape the taxonomic and phylogenetic diversity patterns of subtropical forest communities.

### 3.5. Implications for Management

Our findings indicate that when a native species becomes overly dominant, its ecological effects resemble those of an exotic invader: it can similarly reshape community assembly through environmental filtering and competitive exclusion, thereby undermining the functional stability of the community [3]. This suggests that ecologists should pay closer attention to the over-expansion of native species and incorporate such cases into the conceptual framework of biological expansions and succession [37]. Notably, our study underscores the indispensable role of the phylogenetic diversity dimension in understanding expansion impacts: conserving biodiversity that encompasses a broad range of evolutionary lineages may enhance a community’s resistance to expansion, whereas expansion-driven phylogenetic loss could diminish the ecosystem’s resilience [38]. Our work provides a scientific basis for managing Moso bamboo-invaded forests and restoring affected communities. Given the significant negative impacts of Moso bamboo expansion on tree-layer diversity and soil nutrients, this issue warrants serious attention from forest management and biodiversity conservation authorities. In areas heavily invaded by Moso bamboo, it may be advisable to implement targeted interventions—for example, reducing Moso bamboo density or selectively cutting Moso bamboo (creating canopy gaps)—to restore understory light conditions and soil fertility, thereby facilitating the regeneration of native trees. At the same time, the expansion-resistant species we identified (such as *Cyclobalanopsis glauca*) can be regarded as “key survivors” that serve as functional support in restoration efforts; these species, by virtue of their ability to coexist with Moso bamboo, could be used in mixed-species plantings or as indicator species to help maintain biodiversity in invaded ecosystems. In contrast, species with very low RI (highly susceptible species) should be prioritized for monitoring and protection, and measures should be taken in Moso bamboo-invaded areas to prevent their populations from collapsing or going locally extinct.

Admittedly, our study has several limitations. First, we examined a spatial invasion gradient in only one region with a relatively limited plot size and number, which may not capture the full range of bamboo invasion effects under different environmental conditions at broader scales. In addition, although we integrated both taxonomic and phylogenetic diversity metrics, our discussion of functional trait diversity and related ecological processes remains limited. For example, we did not explore how various species’ functional traits (such as shade tolerance or root competi-tiveness) might influence their invasion resistance. Finally, while we treated the soil-factor changes associated with bamboo expansion as concomitant effects of invasion, the causal relationships between them are still unclear. For instance, it remains uncertain to what extent the observed reductions in soil nutrients and increased pH were a consequence of bamboo invasion, or conversely, whether those soil changes further facilitated bamboo’s competitive dominance. These limitations may affect the interpretation of our results to some degree, but they do not undermine our overall conclusion that bamboo invasion dramatically reshapes community diversity; rather, they highlight directions for future improvement.

Looking ahead, we recommend the following research to extend and build upon our conclusions and their applications: (1) conduct long-term in situ monitoring and controlled experiments to track the dynamic impacts of Moso bamboo expansion on community diversity and soil conditions, verify the inferences drawn from the space-for-time substitution, and assess the ecological outcomes of intervention measures (e.g., removing Moso bamboo or selectively thinning Moso bamboo stands); (2) integrate functional trait and phylogenetic approaches to investigate the biological mechanisms of species’ expansion resistance—for example, by comparing the trait profiles of high-RI (resistant) vs. low-RI (susceptible) species to identify key traits that determine species’ fates under expansion; (3) expand the research scope to other ecosystems and species to test the generality of our findings—for instance, by examining the ecological impacts of other overabundant native vines, shrubs, or trees—thereby enriching expansion ecology theory. Pursuing such studies will further deepen our understanding of the relationships between expansions and community diversity and provide a stronger scientific basis for developing effective biodiversity conservation and ecosystem restoration strategies.

## 4. Materials and Methods

### 4.1. Study Area

The study was conducted in Xikou Town, Longyou County, Quzhou City, Zhejiang Province, China (28°53′ N, 119°14′ E), within a subtropical evergreen broadleaf forest region. The climate of the area is subtropical monsoonal, with a mean annual temperature of about 17.3 °C and mean annual precipitation around 1800 mm. The region has a frost-free period of approximately 230 days per year and about 1600 h of annual sunshine, indicative of a humid climate. The soils in the study area are predominantly acidic Acrisols. The native vegetation is dominated by subtropical evergreen broad-leaved forest, and Moso bamboo plantations are also common in the landscape. In the 1960s–1970s, many native forests in this region were cleared and converted to Moso bamboo stands due to the high economic value of Moso bamboo products. In recent years, however, rising management costs and falling market prices for Moso bamboo have led to the abandonment of some Moso bamboo plantations, which has facilitated the expansion of Moso bamboo into adjacent evergreen broadleaf forests. This context provides an ideal setting to examine the ecological impacts of Moso bamboo expansion on forest community diversity and soil properties.

**Figure 6 plants-14-03231-f006:**
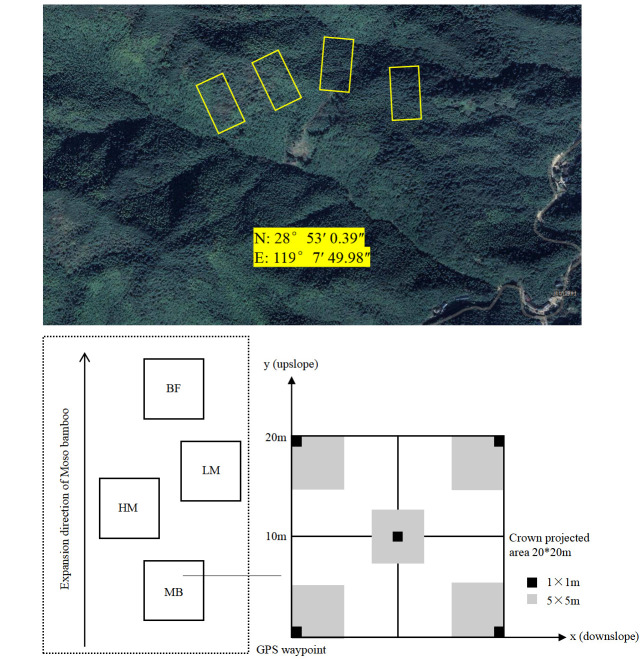
Schematic map of sampling plots. BF, bamboo-free forest; LM, low Moso bamboo; HM, high Moso bamboo; MB, Moso bamboo monoculture.

### 4.2. Plot Design and Sampling

We employed a space-for-time substitution design to represent different stages of Moso bamboo expansion [39]. Four forest types along a Moso bamboo expansion gradient were selected: (1) BF, Moso bamboo-free evergreen broadleaf forest (~0% Moso bamboo stems in the tree layer), representing native forest with no Moso bamboo; (2) LM, low Moso bamboo mixture (20–40% of tree-layer stems are Moso bamboo), representing an early expansion stage; (3) HM, high Moso bamboo mixture (60–80% of tree-layer stems are Moso bamboo), representing a late expansion stage where Moso bamboo is dominant but some broadleaf trees remain; and (4) MB, Moso bamboo forest (~100% Moso bamboo, a virtually pure Moso bamboo stand), representing the endpoint of complete Moso bamboo takeover. Each expansion stage had 4 independent 20 m × 20 m plots (replicates), totaling 16 plots (Figure 6). All plots were located in close proximity with similar topography, soil type, and microclimate, minimizing external environmental differences. The percentage of Moso bamboo cover in the tree layer (i.e., the proportion of Moso bamboo stems) was used to quantify Moso bamboo expansion intensity in each plot. The threshold values (~20% and ~70% Moso bamboo) for defining “low” vs. “high” expansion were based on prior expansion studies, which suggest that when an invasive species reaches ~20% abundance it marks a critical point for ecological impact, and native diversity drops sharply when the invader exceeds ~60–80% dominance [9,40]. This experimental design captures a clear sequence of Moso bamboo expansion from absence to complete dominance.

Within each 20 m × 20 m plot, a stratified vegetation survey was carried out to record plant species composition and structure across tree, shrub, and herb layers. Five 5 m × 5 m subplots were established in each plot (one at each corner and one at the center) to survey the shrub layer, and within each 5 m × 5 m subplot, five 1 m × 1 m quadrats were nested to survey the herb layer. In each plot, all woody individuals with diameter at breast height (DBH) ≥ 2 cm were recorded as tree-layer species, while woody stems with DBH < 2 cm were recorded in the shrub layer. We identified every vascular plant species in each plot and recorded its occurrence and abundance in the respective layer. For each tree-layer individual, we measured species identity, DBH, and height (and crown width for canopy trees); for the shrub layer, we recorded species, number of individuals (or clumps), height, and estimated percent cover within each 5 m × 5 m subplot; for the herb layer, we recorded species, number of individuals (or ramets) or percent cover, and height in each 1 m × 1 m quadrat. If a species occurred in both tree and shrub layers, its occurrences were noted separately in each layer. These detailed field measurements provided the basis for quantifying each species’ dominance in the community (Appendix A).

In each plot, soil samples were collected to assess soil physical and chemical properties. To obtain a representative soil sample, we took 10 soil cores (0–20 cm depth) per plot using a soil auger, following a stratified “2-3-2-3” point arrangement across the plot. The 10 subsamples from a plot were thoroughly mixed to form one composite sample for that plot. Each composite sample was air-dried (after removing stones and plant debris), then gently crushed and passed through a 2 mm sieve for chemical analysis. We measured soil pH, total organic carbon (TOC), total nitrogen (TN), and total phosphorus (TP) for each composite sample using standard protocols. Soil pH was determined by the potentiometric method in a 1:2.5 (soil: water) suspension. Soil TOC was measured by the dichromate oxidation method with heated concentrated sulfuric acid (Walkley–Black procedure). Soil TN was determined by the Kjeldahl digestion method. Soil TP was measured by the molybdenum–antimony colorimetric method after acid digestion [41]. All soil analyses were conducted in the laboratory with appropriate quality controls.

### 4.3. Species Importance Value

After completing the field survey, we compiled a species list for each plot and calculated the importance value (IV) of each species in that plot. The importance value is a composite measure of a species’ relative importance in the community. For each species, we combined its relative abundance, relative dominance (e.g., basal area or cover), and relative frequency in the plot to calculate its IV. This index allowed us to identify which species were dominant or declining across the Moso bamboo expansion gradient. Each species’ IV in each plot was used in subsequent analyses to quantify changes in species prominence along the Moso bamboo expansion gradient [42].
(1)$$Relative\;Density\;\;=\;\frac{Number\;of\;individuals\;of\;a\;given\;species\;in\;the\;plot\;}{Total\;number\;of\;individuals\;of\;all\;species\;in\;the\;plot}\;\times \;100\%$$
(2)$$Relative\;Dominance\;=\;\frac{Basal\;area\;of\;a\;given\;species\;in\;the\;plot\;}{Total\;basal\;area\;\;of\;all\;species\;in\;the\;plot}\;\times \;100\%$$
(3)$$Relative\;Frequency\;\;=\;\frac{Frequency\;of\;occurrence\;of\;a\;given\;species}{Sum\;of\;frequencies\;of\;all\;species\;}\;\times \;100\%$$
(4)$$Importance\;Value\;(IV)\;=\;\frac{Relative\;Density\;+\;Relative\;Dominance\;+\;Relative\;Frequency\;\;}{3}$$

### 4.4. Phylogenetic Tree Construction

We constructed a community phylogenetic tree to evaluate the phylogenetic diversity and relatedness of the plant species (Appendix A). All recorded species (angiosperms) were first standardized according to the Angiosperm Phylogeny Group IV classification system for consistent taxonomy [43]. We then used the R package V.PhyloMaker2 to generate a phylogenetic hypothesis for the species pool. V.PhyloMaker2 attaches the species from our list onto a mega-phylogeny of vascular plants (based on the GBOTB.extended tree) and adds missing taxa following a standardized protocol [44]. This resulted in a fully resolved phylogenetic tree including all species recorded in our plots, with branch lengths inherited from the backbone tree. Moso bamboo was included in the phylogeny, allowing us to quantify the evolutionary distances between Moso bamboo and all other species. The constructed phylogenetic tree was then used for subsequent calculations of phylogenetic diversity metrics.

### 4.5. Diversity Index Calculations

We quantified both traditional taxonomic diversity and phylogenetic diversity for each plot to assess how Moso bamboo expansion affected plant community diversity. For α-diversity, we calculated several standard indices: species richness (number of species), Margalef’s richness index, the Shannon–Wiener diversity index (H′), Simpson’s diversity index (D), and Pielou’s evenness index (J). These indices were computed using standard formulas. Together, they describe changes in species richness, abundance distribution, and community evenness across the expansion gradient. For phylogenetic diversity, we calculated several metrics using the community phylogenetic tree. Faith’s phylogenetic diversity (PD) was computed for each plot as the total branch length spanned by all species in that plot’s subtree (i.e., the cumulative evolutionary history of the community) [13]. We also calculated the mean pairwise phylogenetic distance (MPD) and mean nearest taxon distance (MNTD) between co-occurring species in each plot, which quantify the average phylogenetic relatedness among all species pairs and among nearest-neighbor species, respectively [12]. To assess the community’s phylogenetic structure, we derived the Net Relatedness Index (NRI) and the Nearest Taxon Index (NTI) for each plot. NRI and NTI are standardized effect-size metrics (z-scores) that compare the observed MPD and MNTD of a community to the distribution of MPD/MNTD in randomly assembled communities (null models) We used the “taxa.labels” null model (randomizing species labels across the phylogeny) and performed at least 9999 randomizations for each calculation to obtain robust estimates of NRI and NTI. An NRI or NTI value significantly greater than zero indicates phylogenetic clustering (co-occurring species are more closely related than expected by chance), whereas a value less than zero indicates phylogenetic overdispersion (co-occurring species are more distantly related than expected by chance). All diversity metrics were computed in R (version 4.1.5) using the package “picante” [45].
(5)$$Margalef\;index\;of\;species\;richness:\;R=\frac{\left(S-1\right)}{\ln N}\;$$
(6)$$Simpson\;index:D=\frac{\sum_{i=1}^{S}n_{i}\left(n_{i}-1\right)}{N\left(N-1\right)}\;$$
(7)$$Shannon-Wiener\;index:H^{\prime }=-\sum_{i=1}^{S}Pi\ln p_{i}\;$$
(8)$$Pielou\;evenness\;index:E=\frac{H^{\prime }}{\ln S}$$ where for different community layers, S is the total number of species in the layer, N is the total number of individuals of all species in the layer, n_i_ is the number of individuals of species i in the layer, and P_i_ is the proportion of individuals of species i relative to the total number of individuals in the layer.
(9)$$\begin{array}{c} PD=\sum_{i=1}^{n}\;L_{i} \end{array}\;$$
(10)$$NRI=-1\;\times \;\frac{MPD_{\mathrm{obs}}-MPD_{\mathrm{null}}}{sd\;(MPD_{\mathrm{null}})}\;$$
(11)$$\begin{array}{c} NTI=-1\times \frac{MNTD_{\mathrm{obs}}-MNTD_{\mathrm{null}}}{sd\;(MNTD_{\mathrm{null}})} \end{array}$$ where n is the number of species in the community, and
$L_{i}$ is the branch length connecting the ith species in the phylogenetic tree.
$MPD_{\mathrm{obs}}$ and
$MNTD_{\mathrm{obs}}$ are the observed mean pairwise distance (MPD) and mean nearest taxon distance (MNTD) between species in the community, respectively.
$MPD_{\mathrm{null}}$ and
$MNTD_{\mathrm{null}}$ represent the expected mean values of MPD and MNTD obtained after 999 randomizations under the null model.
$sd\left(MPD_{\mathrm{null}}\right)$ and
$sd\left(MNTD_{\mathrm{null}}\right)$ denote the standard deviations of the randomized MPD and MNTD values, respectively.

### 4.6. Resistance Index (RI)

Grounded in density–response curves and a per-unit-effect framework, we constructed a species-specific Resistance Index (RI) that integrates each species’ change in importance with its phylogenetic distance to Moso bamboo to quantify resistance or susceptibility [46,47,48,49]. For every plant species recorded in our plots, we quantified how its IV responded to increasing Moso bamboo dominance, and we accounted for the species’ evolutionary relatedness to Moso bamboo. First, we assessed the relationship between the species’ IV and the degree of Moso bamboo expansion across the 16 study plots. We treated Moso bamboo expansion intensity (measured as the Moso bamboo canopy cover percentage in each plot; see Statistical Analysis below) as the predictor and the species’ IV as the response. For each species occurring in at least 3 of the 4 expansion stages (to ensure a sufficient data series), we calculated the slope of IV vs. Moso bamboo cover across the plots. Rather than using a simple linear regression (which can be sensitive to outliers given the small number of points), we utilized a Theil–Sen estimator to obtain a robust slope (β) for the change in that species’ IV per unit increase in Moso bamboo cover (Moso bamboo cover was scaled 0 to 1). This slope
$\beta_{i}$ represents the species’ trend: a negative β indicates the species’ IV declines as Moso bamboo cover increases, while a positive β indicates the species becomes more important in plots with higher Moso bamboo cover. Next, we determined the phylogenetic distance between each species *i* and Moso bamboo. Using the community phylogeny, we extracted the total branch length separating species *i* from *p. edulis*. This distance is effectively equivalent to the mean pairwise phylogenetic distance between the species and Moso bamboo (since we are comparing each species to a single focal species on the tree). Finally, we calculated the species’ Resistance Index as the slope divided by this phylogenetic distance:
(12)$$\begin{array}{c} RI_{i}=\frac{\beta_{i}}{D\left(i,\mathrm{Moso}\;\mathrm{bamboo}\right)} \end{array}\;$$ where
$\beta_{i}$ is the Theil–Sen slope for species i’s IV ~ Moso bamboo cover, and *D* (*i*, Moso bamboo) is the phylogenetic distance between species i and Moso bamboo. Thus, a more negative RI indicates a sharper decline in importance for a given evolutionary proximity–in other words, a species that both declines strongly and is closely related to Moso bamboo will have a particularly large negative RI. Conversely, a species with a less negative or positive RI would be relatively “resistant”—either it does not decline much, or it actually increases in IV, especially if it is a close relative of Moso bamboo (a positive β divided by a small distance yields a high positive RI). By comparing RI values among species, we can identify which species are most vulnerable to being displaced by Moso bamboo and which are relatively tolerant or even favored under Moso bamboo expansion. After computing raw RI values for all species in a given layer, we standardized these values (z-score within that layer) to facilitate comparisons across layers with different diversities. We operationally defined Resistant species as those with a standardized RI ≥ +1 (i.e., at least one standard deviation above the mean, indicating a strong positive response to expansion), Susceptible species as those with RI ≤ –1 (one standard deviation below the mean, indicating a strong negative response), and Neutral species as those with intermediate RI (|RI| < 1).

### 4.7. Statistical Analysis

We used nonparametric statistical tests to examine the effects of Moso bamboo expansion on community soil properties, species diversity, and phylogenetic diversity indices. Specifically, the Kruskal–Wallis test was employed to assess overall differences among expansion stages (α = 0.05), followed by Dunn’s test for post hoc multiple comparisons. To describe monotonic trends of diversity and soil properties along the expansion intensity gradient (scaled 0–1), we applied Theil–Sen slope estimation combined with Mann–Kendall trend tests [50]. The Benjamini–Hochberg false discovery rate (BH-FDR) correction was used to control the expected proportion of false positives at α = 0.05 [51]. These nonparametric procedures do not assume normally distributed residuals, are robust to outliers and heteroscedasticity, and are well-suited for small sample sizes. (Although the Mann–Kendall test is often applied to time series, it is a rank-based test of monotonic association and does not require temporal ordering; here we applied it to a continuous spatial gradient of Moso bamboo cover.)

To quantify the variance in community diversity indices explained by Moso bamboo cover versus soil variables, we conducted redundancy analysis (RDA) [52]. Separate RDA models were constructed for the tree, shrub, and herb layers. We selected the Shannon–Wiener index, Faith’s phylogenetic diversity (PD), and the phylogenetic structure metric NTI as response variables because this combination captures complementary dimensions of community structure (taxonomic, basal phylogenetic, and tip phylogenetic) while avoiding redundancy between PD and MPD-based metrics (e.g., NRI), thereby preserving ecological interpretability and model robustness. Moso bamboo cover and soil factors were used as explanatory variables. Model significance was assessed by ≥999 permutation tests, and explanatory power was reported as adjusted R^2^. Variance partitioning analysis (VPA) was further used to disentangle the proportions of variance explained uniquely by Moso bamboo cover, uniquely by soil variables, and jointly by both [53]. Prior to multivariate regression and constrained ordination, we calculated variance inflation factors (VIF) for all environmental predictors; all VIF values were < 5, indicating no severe multicollinearity and thus justifying the simultaneous inclusion of Moso bamboo cover and soil variables.

All data processing, statistical analyses, and visualization were conducted in R version 4.1.5. Statistical significance was set at α = 0.05. Results are reported as means ± standard errors (SEs), and significant differences are indicated in figures using error bars or distinct letter annotations.

## 5. Conclusions

Our study shows that along a clear expansion gradient, overstory tree diversity and evolutionary breadth contracted markedly, while the understory (particularly herbs) exhibited gains in species richness with partial increases in phylogenetic breadth. These biotic shifts co-occurred with consistent edaphic reconfiguration (soil pH rising, TOC and TN declining), and community reassembly appears to be driven primarily by a composite “Moso bamboo × soil” factor rather than by stand structure alone. In summary, we demonstrated how the massive expansion of Moso bamboo, a native dominant species, undermines forest biodiversity at both the species and phylogenetic levels and alters community assembly processes, thereby deepening our understanding of forest expansion ecology. We applied a phylogenetically informed Resistance Index (RI), grounded in existing density–response frameworks, to classify each species’ response to bamboo expansion. This RI successfully distinguished canopy “losers” from understory “winners”, consistent with patterns of competitive exclusion and environmental filtering. Although useful, this index should be viewed as a preliminary tool requiring further validation. In terms of management, our findings highlight specific actions. We recommend targeted reduction in bamboo density (e.g., selective thinning) and restoration of soil conditions (e.g., adding organic amendments to replenish nutrients and adjusting pH) to facilitate native tree regeneration. We also suggest enrichment planting or mixed-species planting of resistant native species (such as *Cyclobalanopsis glauca*, *Ardisia japonica*, *Rubus buergeri*) to support community recovery. In contrast, highly susceptible species (e.g., *Schima superba*, *Symplocos stellaris*) should be monitored closely and protected to prevent local extirpation. Overall, addressing the spread of even native “invaders” like Moso bamboo requires integrated management (including long-term biodiversity monitoring) to maintain forest diversity.

## Figures and Tables

**Figure 1 plants-14-03231-f001:**
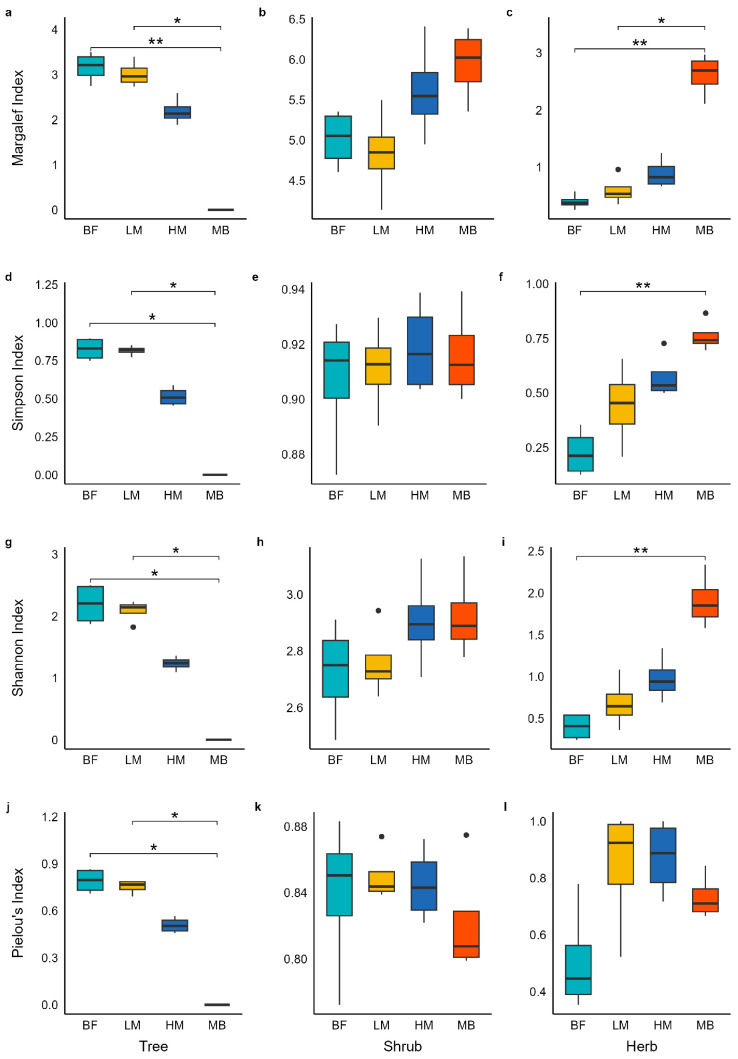
Changes in species diversity across the Moso bamboo expansion gradient for tree, shrub, and herb layers. BF, bamboo-free forest; LM, low Moso bamboo; HM, high Moso bamboo; MB, Moso bamboo monoculture. Boxplots show the median and interquartile range. Boxplots show the median and interquartile range (*N* = 4, * *p* ≤ 0.05, ** *p* ≤ 0.01, Dunn’s test).

**Figure 2 plants-14-03231-f002:**
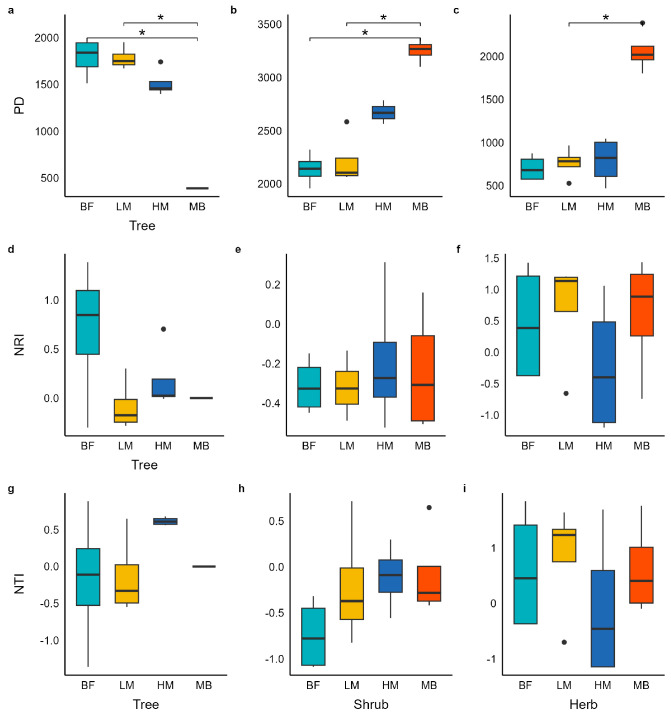
Changes in phylogenetic diversity across the Moso bamboo expansion gradient for the tree, shrub, and herb layers. Different lowercase letters denote significant differences among stages at *p* ≤ 0.05. BF, bamboo-free forest; LM, low Moso bamboo; HM, high Moso bamboo; MB, Moso bamboo monoculture. Boxplots show the median and interquartile range. Boxplots show the median and interquartile range (*N* = 4, * *p* ≤ 0.05, Dunn’s test).

**Figure 3 plants-14-03231-f003:**
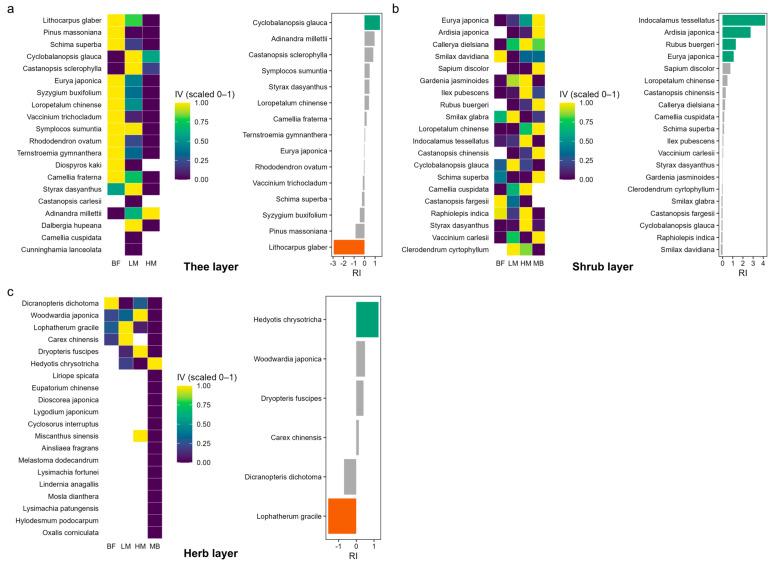
Species-level changes in dominance and expansion resistance across the tree, shrub, and herb layers under Moso bamboo expansion. In each panel (**a**–**c**), the left heatmap shows species importance values (IV) for the top 20 native species, with row-wise 0–1 normalization; warmer colors indicate higher relative dominance of that species at a given stage, cooler colors indicate lower. The right bar plot shows the standardized Resistance Index (RIz) for the same species: green = Resistant (RIz ≥ +1), orange = Susceptible (RIz ≤ −1), grey = Neutral. Positive values indicate rising importance with expansion after accounting for phylogenetic distance to Moso bamboo, whereas negative values indicate decline. Note that heatmap colors are not comparable across species (row-normalized) and should be interpreted as within-species trends.

**Figure 4 plants-14-03231-f004:**
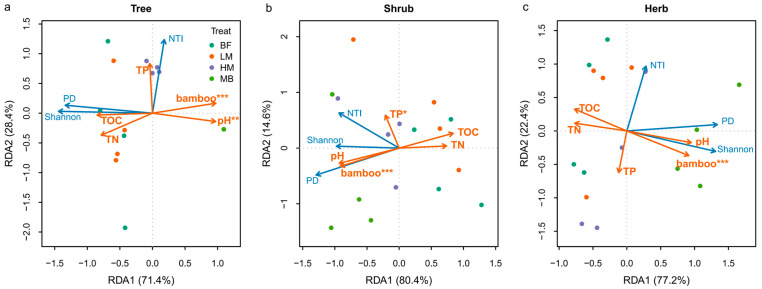
RDA ordination of community diversity indices constrained by the Moso bamboo–soil environmental gradient. (**a**) Tree layer; (**b**) Shrub layer; (**c**) Herb layer. Orange arrows denote explanatory variables (Moso bamboo cover and soil factors), and blue arrows denote response variables (Shannon–Wiener diversity, Faith’s PD, and NTI). Arrow direction indicates increasing values of the corresponding variable, and arrow length reflects the strength of its correlation with the ordination axes (longer = stronger). Points represent plots. Axes (RDA1, RDA2) show the percentage of constrained variance explained. Asterisks mark the significance of environmental variables based on permutation tests (* *p* < 0.05, ** *p* < 0.01, *** *p* < 0.001).

**Figure 5 plants-14-03231-f005:**
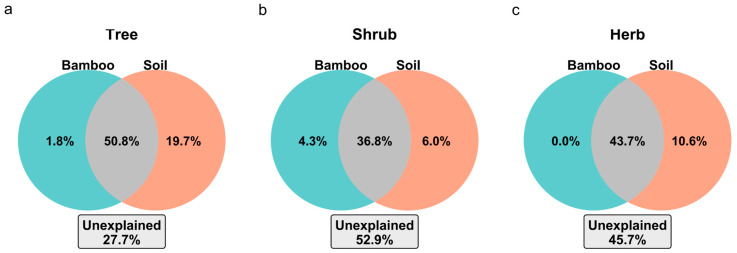
VPA of the variance in community diversity indices explained by Moso bamboo cover and soil variables. The bar graphs show the percentage of variation in diversity indices (for each vegetation layer) that is uniquely attributable to Moso bamboo cover alone (green), uniquely to soil factors alone (brown), and jointly to the combined Moso bamboo–soil factors (gray).

**Table 1 plants-14-03231-t001:** Soil physicochemical properties across different stages of Moso bamboo expansion.

Factor	BF	LM	HM	MB	τ	*p*
pH	4.39 ± 0.16 a	4.62 ± 0.14 ab	4.76 ± 0.12 b	5.24 ± 0.1 c	0.85	<0.001 ***
TOC (g/kg)	30.74 ± 3.49 c	24.78 ± 2.11 b	19.47 ± 0.66 a	17.07 ± 1.62 a	−0.791	<0.001 ***
TN (g/kg)	2.13 ± 0.28 c	1.95 ± 0.11 bc	1.61 ± 0.13 ab	1.48 ± 0.19 a	−0.668	0.01 **
TP (g/kg)	0.22 ± 0.07	0.23 ± 0.06	0.23 ± 0.03	0.23 ± 0.02	0.158	0.569

BF, bamboo-free forest; LM, low Moso bamboo; HM, high Moso bamboo; MB, Moso bamboo monoculture; TOC, total organic carbon; TN, total nitrogen; TP, total phosphorus. Different lowercase letters in the same rows indicate significant difference at *p* ≤ 0.05 (Dunn’s test, mean ± SD, *N* = 4). τ, Kendall’s rank correlation coefficient for trend vs. bamboo cover; *p*, significance of trend (** *p* ≤ 0.01, *** *p* ≤ 0.001).

**Table 2 plants-14-03231-t002:** Species diversity indices in the tree, shrub, and herb layers along the Moso bamboo expansion gradient.

Layer	Factor	BF	LM	HM	MB	τ	*p*
Tree	Margalef	3.17 ± 0.34 a	3.02 ± 0.29 a	2.19 ± 0.3 b	0 c	−0.775	<0.001 ***
	Pielou	0.79 ± 0.08 a	0.75 ± 0.05 a	0.51 ± 0.05 b	0 c	−0.793	<0.001 ***
	Simpson	0.83 ± 0.08 a	0.82 ± 0.03 a	0.51 ± 0.06 b	0 c	−0.757	<0.001 ***
	Shannon	2.2 ± 0.34 a	2.09 ± 0.18 a	1.23 ± 0.11 b	0 c	−0.793	<0.001 ***
	PD	1796.64 ± 216.36 a	1781.46 ± 121.94 a	1514.07 ± 153.99 ab	390.7 ± 0 b	−0.649	0.002 **
	NTI	0.77 ± 0.75	−0.05 ± 0.28	0.2 ± 0.37	0 ± 0	0.072	0.800
	NRI	−0.16 ± 0.92	−0.08 ± 0.54	0.71 ± 0.07	0 ± 0	−0.162	0.569
Shrub	Margalef	5.02 ± 0.36 ab	4.84 ± 0.56 b	5.62 ± 0.61 ab	5.95 ± 0.45 a	0.527	0.0123 *
	Pielou	0.84 ± 0.05	0.85 ± 0.01	0.84 ± 0.02	0.82 ± 0.03	−0.246	0.308
	Simpson	0.91 ± 0.03	0.91 ± 0.02	0.92 ± 0.02	0.92 ± 0.02	0.088	0.756
	Shannon	2.73 ± 0.18	2.76 ± 0.13	2.91 ± 0.17	2.93 ± 0.15	0.369	0.095
	PD	2140.3 ± 150.22 b	2214.21 ± 247.32 b	2671.78 ± 95.96 ab	3276.66 ± 113.42 a	0.720	<0.001 ***
	NTI	−0.27 ± 0.16	−0.29 ± 0.18	−0.13 ± 0.39	−0.27 ± 0.3	0.369	0.095
	NRI	−0.69 ± 0.4	−0.22 ± 0.65	−0.07 ± 0.32	−0.13 ± 0.51	0.105	0.709
Herb	Margalef	0.4 ± 0.13 b	0.6 ± 0.26 b	0.89 ± 0.26 b	2.57 ± 0.33 a	0.703	1
	Pielou	0.51 ± 0.19 b	0.84 ± 0.22 ab	0.88 ± 0.13 a	0.74 ± 0.08 ab	0.132	0.625
	Simpson	0.23 ± 0.11 c	0.45 ± 0.19 bc	0.58 ± 0.11 ab	0.76 ± 0.07 a	0.750	<0.001 ***
	Shannon	0.4 ± 0.16 c	0.68 ± 0.3 bc	0.98 ± 0.27 b	1.9 ± 0.31 a	0.767	<0.001 ***
	PD	701.96 ± 150.85 b	763.89 ± 180.7 b	788.89 ± 274.74 b	2034.17 ± 200.47 a	0.545	0.011 *
	NTI	0.45 ± 0.96	0.68 ± 0.94	−0.23 ± 1.06	0.6 ± 0.98	−0.018	0.996
	NRI	0.55 ± 1.15	0.81 ± 1.01	−0.11 ± 1.37	0.62 ± 0.78	0	1

BF, bamboo-free forest; LM, low Moso bamboo; HM, high Moso bamboo; MB, Moso bamboo monoculture; TOC, total organic carbon; TN, total nitrogen; TP, total phosphorus. Different lowercase letters in the same rows indicate significant difference at *p* ≤ 0.05 (Dunn’s test, mean ± SD, *N* = 4). τ, Kendall’s rank correlation coefficient for trend vs. bamboo cover; *p*, significance of trend ( * *p* ≤ 0.05, ** *p* ≤ 0.01, *** *p* ≤ 0.001).

## Data Availability

The datasets in this study are included in the article and Appendix A; further inquiries for materials should be directed to LingFeng Mao.

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
