# Peer review of "The Impact of Phyllostachys heterocyclas Expansion on the Phylogenetic Diversity and Community Assembly of Subtropical Forest"

_plants, 2025, doi:10.3390/plants14203231_

Round 1
Reviewer 1 Report
Comments and Suggestions for Authors
The article is devoted to current and important issue of expansion of species into forests. The authors studied the expansion of Phyllostachys heterocyclas influences species diversity and phylogenetic diversity across subtropical forest strata (trees, shrubs, herbs) as well as soil properties. The authors use methods adequate to the tasks. The study has theoretical and practical significance for forestry.
Title
The title is appropriate to the content, but in my opinion, it needs editing. Perhaps it should be formulated more generally. For example, "The Impact of Phyllostachys heterocyclas Expansion on the Phylogenetic Diversity and Community Assembly of Subtropical Forest"
Abstract
This section allows you to evaluate the significance of the study, its purpose, the methods used, and the results obtained. I recommend that authors indicate the region of the study and supplement the abstract with quantitative results.
Introduction
The introduction contains enough necessary information to understand the research. The references provided are relevant to the topic of the study. However, there are quite a lot of references older than 5 years.
Materials and Methods
In my opinion, it's not very convenient that the objects and methods are placed at the end of the article. It's better to place them after the introduction.
The study area is not described in sufficient detail. I recommend providing a map of the study area and supporting this section with references. It would also be better to provide the soil names according to the international classification (WRB, FAO).
Subsection 4.2 identifies four forest types. What typology are they based on?
The methods and approaches used must be accompanied by references.
Was soil morphology studied?
Perhaps it would be helpful to present the vegetation and soil parameters studied in a table, indicating the methods used and providing references.
Subsection 4.3 should be supported with references.
Results
The results are presented quite clearly and illustrated with 5 figures and 2 tables. However, there are a few comments.
Table 1. Add the meaning of the symbols in the note to the table: BF, LM, HM, MB.
Figure 1. Add the meanings of the symbols to the figure captions: BF, LM, HM, MB.
Table 2. Add the meaning of the symbols in the note to the table: BF, LM, HM, MB. Remove the symbols (TOC, TN, TP) from the table explanation, as they are not in the table.
Figure 2, 4. All symbols must be explained in the figure caption.
Discussion
The authors have discussed the results of the work quite well.
Conclusions
The conclusions are consistent with the results, but rather brief. Perhaps they should be expanded and supported by the most important quantitative results. Also, the conclusions do not include information on which species are most vulnerable to bamboo expansion.
Reviewer 2 Report
Comments and Suggestions for Authors
The manuscript entitled “Moso Bamboo (Phyllostachys heterocyclas) expansion drives loss of tree phylogenetic diversity and reorganizes community assembly” provides an integrative analysis of taxonomic and phylogenetic diversity changes under Moso bamboo expansion. The study combines vegetation surveys, soil parameters, and a novel Resistance Index (RI), offering valuable insights into the ecological consequences of native species overexpansion. The manuscript is overall well-written and scientifically relevant, but several aspects require revision to strengthen its rigor and applicability to forest management.
Introduction
The introduction provides a solid background, situating Moso bamboo expansion within the broader framework of biodiversity loss and “native expansions.” The rationale for considering phylogenetic diversity is appropriate. However, the review of existing literature is somewhat repetitive (lines 54–70, 83–89). A more concise synthesis would help emphasize the specific research gaps. The main research questions are well introduced, but explicit hypotheses remain vague (lines 105–112). Formulating clear, testable hypotheses would strengthen the scientific framing.
Methods
The plot design, sampling protocol, and soil analyses are described in detail, ensuring reproducibility. The introduction of a Resistance Index (RI) is a methodological innovation. However, the reliance on a space-for-time substitution (lines 471–476) constrains inference on long-term dynamics. The limitations of this design should be acknowledged more explicitly. The total of 16 plots (4 stages × 4 replicates) may limit statistical generalizability. This caveat should be highlighted. The RI calculation (lines 576–616) is technically thorough but overly dense. A schematic or flow diagram would improve accessibility for readers.
Results
Results are systematically presented across forest strata (trees, shrubs, herbs). The decline in canopy diversity versus increases in understory diversity is clearly shown. However, Table 2 (lines 151–154) and Figure 1 (lines 148–149) both display diversity indices, leading to redundancy. Simplifying presentation would improve clarity. Some sentences include discussion-like interpretations (e.g., lines 232–239: “greater phylogenetic breadth but reassembly”), which should be moved to the Discussion section. Figure 3 presents too many species simultaneously, obscuring the key message. Dividing into canopy vs. understory, or moving secondary species to the Supplementary section, would be better.
Discussion (lines 265–452)
The discussion highlights differences across vegetation strata, emphasizes soil-mediated effects, and introduces the significance of phylogenetic diversity. However, many findings are restated almost verbatim from the Results (e.g., tree diversity decline, lines 278–287 vs. 131–134). This reduces analytical depth. Although soil effects are discussed (lines 387–399), other mechanisms such as allelopathy, root competition, or microbial shifts are not sufficiently addressed. Suggested implication (lines 410–437) are limited to thinning or cutting bamboo. Broader recommendations (e.g., soil restoration, mixed-species planting, monitoring strategies) would enhance practical management. The discussion lacks cross-regional or cross-species comparisons. Relating bamboo expansion to other cases of “native invaders” would strengthen generalizability.
Conclusion
The conclusion effectively synthesizes the study’s findings on biodiversity loss under bamboo expansion and introduces the Resistance Index as a promising yet preliminary tool, but it overstates its novelty and provides overly general management recommendations without sufficiently actionable guidance.
Reviewer 3 Report
Comments and Suggestions for Authors
This is an excellent and interesting paper on Moso bamboo impact on broadleaf subtropical forests. I particularly liked the inclusion of the phylogenetic diversity impact. This is often not done and the authors make a sound case for its inclusion.
The only error I noted was in Table 1. The p values for Total Nitrogen and Total Phosphorus are mixed up. I believe they should be reversed to be in agreement with the text.
The English is excellent and I could not detect any errors in usage, grammar, spelling, or punctuation.
Author Response
Dear reviewers, thank you very much for taking the time to review our manuscript. We thank the reviewer for this helpful reminder.The order of τ and p values in the BH-FDR results of Table 1 was incorrect and has now been corrected in the manuscript.
Reviewer 4 Report
Comments and Suggestions for Authors
The manuscript “Moso Bamboo (Phyllostachys heterocyclas) expansion drives loss of tree phylogenetic diversity and reorganizes community assembly” (authors: Jiannan Wang, Ru Li, Zichen Huang, Sili Peng, Zhiwei Ge, Xiaoyue Lin, Lingfeng Mao) aims to determine how the expansion of Moso bamboo influences both species diversity and phylogenetic diversity across forest community strata (trees, shrubs, and herbs). Despite the fact that the research is of some value, because it helps to advance the understanding of the mechanisms of “native expansion”, the text of the MS contains many flaws. However, let's take it in order, according to the lines:
125-126: The contents of Table 1 are puzzling, namely the lines:
TN (g/kg) 2.13 ± 0.28c 1.95 ± 0.11bc 1.61 ± 0.13ab 1.48 ± 0.19a 0.158 0.569
and
TP (g/kg) 0.22 ± 0.07 0.23 ± 0.06 0.23 ± 0.03 0.23 ± 0.02 0.85 <0.001***
Obviously, there are here errors in calculations.
149: In Figure 1, the caption says: “Different lowercase letters denote significant differences among stages at p ≤ 0.05”. However, the letters in the Figure indicate the variants of the 4 indexes for tree, shrub, and herb layers (by the way, why this was done is unclear, since they are not mentioned both in the caption and in the MS text), and the statistical differences are represented as lines with asterisks, but the differences using lowercase letters are highlighted in Table 2. And yet, in this regard, there is no consistency between Figure 1 and Table 2 regarding the allocation of statistical differences between the variants.
174-175: In Figure 2, the caption also says “Different lowercase letters denote significant differences among stages at p 174 ≤ 0.05”. But the letters here indicate the variants of 3 indexes for tree, shrub, and herb layers (by the way, why these letter indexes are highlighted is unclear, because these indexes are not mentioned in the text of the MS). The statistical differences in the figure are shown as lines with asterisks.
190: In Figure 3, the maximum values in IV (Scaled 0-1) are marked in yellow, and slightly lower values are marked in green. But on the Thee layer graph (obviously, the Tree layer) Cyclobalanopsis glauca is placed above Adinandra millettii, although the first species is marked in green and the second in yellow.
193-194: Although it is reported that in Figure 3 “Orange indicates Susceptible species, and blue indicates Resistant species”, in fact, there is obvious confusion with the display of colors.
470: Sampling instead of Sampling
Between 526 and 527: In the denominator of formula (3), the expression “Sum of frequencies of all species” will not this be equal to 100%?
Between 564 and 565: argalef instead of Margalef
577-579: The words “we developed” in the phrase: “Grounded in density–response curves and a per-unit-effect framework, we developed a species-specific Resistance Index (RI) that integrates each species’ change in importance with its phylogenetic distance to Moso bamboo to quantify resistance or susceptibility [41-44]” raise questions.
A general question arises for Table 2 and Figures 1 and 2. It remains unclear why the variation across the Moso bamboo expansion gradient (BF, LM, HM, and MB) for tree, shrub, and herb layers is presented in the MS for 5 indexes (Margalef, Pielou, Simpson, Shannon, and PD) both in tabular and graphical form, whereas for the other two indexes (NRI and NTI) only in graphical form. In other words, the estimates of the variation of the mean and SE for NRI and NTI are not given in the MS.
For some reason, in Part 4. “Materials and Methods” (obviously in 4.7 Statistical Analysis), there is no information about which gradations were displayed on the box-and-whisker plots (quartiles, median, or average) in Figures 1 and 2.
In tables S1, the names of plant species are either italicized or given in plain font.
Round 2
Reviewer 2 Report
Comments and Suggestions for Authors
The authors have revised the manuscript in accordance with the reviewer’s comments, and the paper has been satisfactorily improved. It is now considered suitable for publication.
Author Response
Thanks for your assessment and suggestions!
Reviewer 4 Report
Comments and Suggestions for Authors
The manuscript “The Impact of Phyllostachys heterocyclas Expansion on the Phylogenetic Diversity and Community Assembly of Subtropical Forest” (formerly “Moso Bamboo (Phyllostachys heterocyclas) expansion drives loss of tree phylogenetic diversity and reorganizes community assembly”, authors: Jiannan Wang, Ru Li, Zichen Huang, Sili Peng, Zhiwei Ge, Xiaoyue Lin and Lingfeng Mao) has been significantly revised by the MS authors and, in my opinion, is now quite ready for publication.
Author Response

(The authors gave the same response as above.)
